# Impact of the COVID-19 pandemic on anxiety and depression symptoms of young people in the global south: evidence from a four-country cohort study

Catherine Porter [1], Marta Favara,[2] Annina Hittmeyer,[2] Douglas Scott,[2] Alan Sánchez Jiménez,[3] Revathi Ellanki,[4] Tassew Woldehanna,[5] Le Thuc Duc,[6] Michelle G Craske,[7] Alan Stein[8,9]

For numbered affiliations see end of article.

**Correspondence to**
Dr Catherine Porter;
catherine.porter@lancaster.ac.uk

## ABSTRACT

**Objective** To provide evidence on the effect of the COVID-19 pandemic on the mental health of young people who grew up in poverty in low/middle-income countries (LMICs).

**Design** A phone survey administered between August and October 2020 to participants of a population-based longitudinal cohort study established in 2002 comprising two cohorts born in 1994–1995 and 2001–2002 in Ethiopia, India (Andhra Pradesh and Telangana), Peru and Vietnam. We use logistic regressions to examine associations between mental health and pandemic-related stressors, structural factors (gender, age), and lifelong protective/risk factors (parent and peer relationship, wealth, long-term health problems, past emotional problems, subjective well-being) measured at younger ages.

**Setting** A geographically diverse, poverty-focused sample, also reaching those without mobile phones or internet access.

**Participants** 10 496 individuals were approached; 9730 participated. Overall, 8988 individuals were included in this study; 4610 (51%) men and 4378 (49%) women. Non-inclusion was due to non-location or missing data.

**Main outcome measures** Symptoms consistent with at least mild anxiety or depression were measured by Generalized Anxiety Disorder-7 (≥5) or Patient Health Questionnaire-8 (≥5).

**Results** Rates of symptoms of at least mild anxiety (depression) were highest in Peru at 41% (32%) (95% CI 38.63% to 43.12%; (29.49–33.74)), and lowest in Vietnam at 9% (9%) (95% CI 8.16% to 10.58%; (8.33–10.77)), mirroring COVID-19 mortality rates. Women were most affected in all countries except Ethiopia. Pandemic-related stressors such as health risks/expenses, economic adversity, food insecurity, and educational or employment disruption were risk factors for anxiety and depression, though showed varying levels of importance across countries. Prior parent/peer relationships were protective factors, while long-term health or emotional problems were risk factors.

**Conclusion** Pandemic-related health, economic and social stress present significant risks to the mental health of young people in LMICs where mental health support is limited, but urgently needed to prevent long-term consequences.

## Strengths and limitations of this study

► The study uses data from adolescents and young adults who grew up in poverty in four low/middle-income countries which were diversely affected by the COVID-19 pandemic, therefore investigating a globally vulnerable but understudied group both in terms of age and wealth.

► This study reaches a broad sample of young people who grew up in poverty, including those without internet or direct access to a mobile phone.

► A key strength is combining a broad range of pandemic-related stressors from survey data on experiences of COVID-19 with previously measured information on longer term risk and protective factors, therefore contributing to a more complete picture of COVID-19 effects.

► A limitation of the study is that it does not have a directly comparable pre-COVID-19 baseline for depression/anxiety, however, proxy variables are used as a baseline and the explanatory variables capture dynamics that happened during the pandemic.

► A further limitation is possible under-reporting due to stigma associated with mental health, despite piloting and validation, as well as possible bias in self-reported experiences of pandemic-related stressors due to feelings of anxiety or depression.

## INTRODUCTION

The COVID-19 pandemic is creating concerns about the mental health of young people around the globe. There has been a call for research funders and researchers to 'deploy resources to understand the psychological effects'[1] of the COVID-19 pandemic and the ensuing 'mental health crisis'.[2] The crisis likely exacerbates previous risk factors of poverty and vulnerability. The Lancet Commission on Global Mental Health had already identified poverty as a key risk factor for the onset and persistence of mental disorders.[3] A recent study[4] found that those with

BMJ

the lowest income were much more likely to suffer from anxiety and depressive disorders than their wealthier counterparts and points to the bidirectional causal relationship between poverty and mental health.

Several studies have examined the mental health impacts of the pandemic, predominantly in high-income countries.[5–7] The few studies from low/middle-income countries (LMICs) have primarily relied on convenience samples and internet-based surveys,[8–11] which are unlikely to reach the rural poor, though one study[12] investigated the effect of immediate lockdown orders on (adult) women's mental health and experiences of intimate partner violence using a phone survey in rural Bangladesh.

Half of all mental health conditions develop by 14 years of age and 75% by early adulthood.[3] In developed countries, young women aged 16–24 years are the most likely to have experienced a deterioration in mental health during the pandemic.[13] Thus, understanding risk and protective factors during the pandemic at this age is critical to prevention, especially for the poorest. There is little research on the mental health of adolescents in LMICs, though they make up the bulk of the global adolescent population.[14]

This study examines the impact of the COVID-19 pandemic on the mental health of nearly 10 000 young people from a 20-year cohort study operating in four LMICs: Ethiopia, India (Andhra Pradesh and Telangana), Peru and Vietnam. When the cohorts were originally recruited, the objective was to ensure that families living in poverty were substantially represented.[15–18] Nowadays, these countries represent a diverse set of experiences during the pandemic, in terms of number and severity of cases, as well as policy responses. Figure 1 shows that COVID-19 has had by far the most striking impact in Peru in terms of deaths per population, followed by India. In

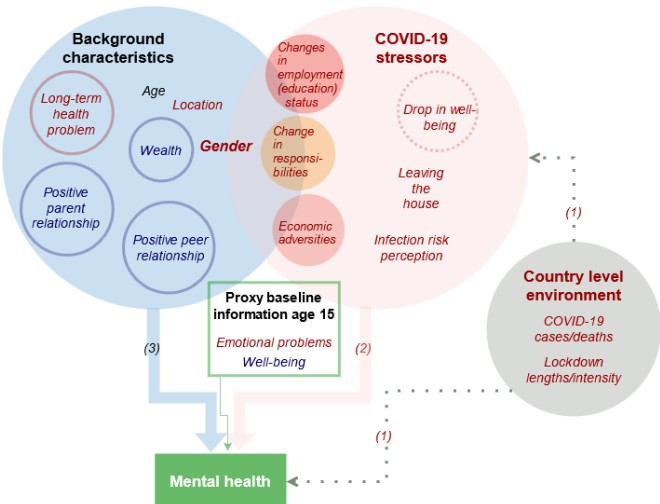

**Figure 2** Theoretical framework of the hypothesised impact of COVID-19 stressors, background characteristics and the country-level environment on mental health. (1), (2) and (3) are the channels as discussed in the framework. Red font colour indicates hypothesised risk factors, blue font colour hypothesised protective factors. In case of gender, women are the hypothesised vulnerable group. Urban participants are hypothesised to be more vulnerable than rural residents (location). Black font colour indicates a potential effect in either direction. White font colour refers to outcome variables. (Solid) lines indicate that the variable was measured in previous in-person rounds. Filled subcircles are categorical/composite variables. Boxes with no fill colour indicate robustness checks to the main framework. Dotted arrows indicate that the variable was only considered implicitly for descriptive statistics. Solid arrows indicate use in logistic regressions. Changes in employment status and changes in the educational status (=educational disruption) are used interchangeably for the younger cohort. Mental health was measured using the GAD-7 and PHQ-8 and a cut-off of five reflecting at least mild symptoms of anxiety/depression was chosen. GAD-7, Generalized Anxiety Disorder-7; PHQ-8, Patient Health Questionnaire-8.

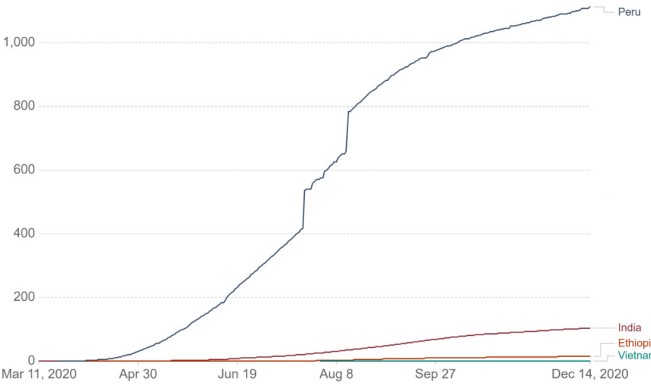

**Figure 1** Cumulative confirmed COVID-19 cases per million people in the four Young Lives countries. Source: Johns Hopkins University CSSE COVID-19 Data,[71] accessed via our world in data.[59] Last updated 15 December 2020. Testing and challenges in the attribution of the cause of death mean that the number of confirmed deaths may not be an accurate count of the true number of deaths from COVID-19. CSSE, Center for Systems Science and Engineering

contrast, Vietnam has been hailed as a success story in controlling the spread of the virus.

Our structural framework is a set of three hypothesised channels through which the pandemic may affect mental health, which are illustrated in figure 2. The first is the country-level environment. Pressure on mental health is likely to be greater in countries which are more affected by the pandemic. Second, within each country, stressors related to changes in circumstances/behaviours/wellbeing that occurred due to the pandemic are hypothesised to negatively affect mental health,[19 20] which we term as *COVID-19-related stressors*. These include individuals' perceived infection risk, economic adversities, changes in employment status and increased household responsibilities, educational disruption, and changes in subjective well-being (SWB) between 2016 and the pandemic. Third, we consider individual, household, and contextual *background characteristics* which may be protective and/or interact negatively with pandemic-related stressors.

We hypothesised that residents of urban areas may have difficulties social distancing in slum-like conditions and are possibly more likely than residents in rural areas to develop mental health conditions. Similarly, depression and anxiety symptoms have been shown to increase more among women than men in other countries.[5 21] Changes imposed by COVID-19 in time use,[22] education and work may also impact the genders differently.[23 24] Further, we exploited information collected in previous survey rounds to investigate characteristics measured during childhood and adolescence: positive social interactions (parent and peer relations), household wealth and long-term health conditions. We also include *proxy baseline information for mental health* in the form of past emotional problems and SWB measured 10 years ago at age 15 years. Our data and methods allow us to discuss the first hypothesis and to directly test the second and third hypotheses.

## METHODS
### Study design and participants
A phone survey[25–28] (see online supplemental documents) was administered between August and October 2020 as part of the Young Lives Study,[29] a longitudinal survey established in 2002 following two cohorts of children born in 1994–1995 and 2000–2001 in Ethiopia, India (Andhra Pradesh and Telangana), Peru and Vietnam. The choice to focus on these four countries was made by the original study team in 2001. The Young Lives Study was designed to monitor the effectiveness of the Millennium Development Goals (2000–2015) in reducing childhood poverty in varied political-economic and sociocultural settings. This led to the selection of four study countries, reflecting a diverse set of geography and development stages[29].

Respondents have been surveyed in person every 3 years, with five consecutive rounds completed by 2016. In 2020, the cohort members are aged 18–19 (younger cohort) and 25–26 years (older cohort). The original sample was selected to include a significant coverage of poorer areas.[15–18] Ninety-three per cent of the cohort (9704) were tracked in 2019. The sample was reduced to 8988 individuals due to missing values for any question (online supplemental tables 1 and 2) including between 0.1% (Vietnam) and 2% (Peru) who did not respond to the mental health questions (online supplemental table 3).

The phone survey was administered over mobile telephone by up to 15 trained interviewers per country, who were provided with the hardware, software and internet access required for working from home. In Ethiopia, roughly 51% of participants did not have access to the internet via a working smartphone or a home computer. Thus, we used local guides who provided sanitised mobile phones for those who did not have them. Responses were recorded in an electronic questionnaire using *Surveybe Implementer* software.

Symptoms of anxiety and depression were measured using the Generalized Anxiety Disorder-7 (GAD-7) scale and the Patient Health Questionnaire depression scale-8 (PHQ-8). The GAD-7 has been validated[30] and used in all four study countries.[31–36] The PHQ-9 was also validated[37–41] and used in several studies.[33–36 42–47] The ninth question of PHQ was dropped due to ethical concerns about how to provide support. The scales were slightly adapted for administration in a phone survey. First, we asked participants whether they were alone in the room and if not, whether they could find a quiet space and/or make sure their phone speaker was off. Second, for each item in GAD-7 and PHQ-8, we asked whether the symptom had been observed (Yes/No) over the past 14 days, and if 'Yes' we then asked about the frequency. The scales were administered as the last section of the survey.

GAD-7 scores between 5 and 9, 10 and 14 and above 15 represent mild, moderate and severe anxiety, respectively.[48] A PHQ-8 score between 5 and 9, 10 and 14, 15 and 19 and above 19 was considered representative of mild, moderate, moderately severe and severe depression, respectively.[49] Cronbach's alpha[50] for both scales was close to or above 0.7.[51] Interitem correlations fell within the recommended range (0.15–0.50[52 53]; online supplemental table 4).

### Statistical analysis
In table 1 we present t-tests to show differences between groups (eg, male/female, urban/rural). Logistic regressions were used to examine the relationship between a range of stressors on a binary variable indicating (at least) mild anxiety (GAD-7 ≥5) and (at least) mild depression (PHQ-8 ≥5), as reported in the Results section. We include two sets of stressors hypothesised to be associated with mental health: changes in circumstances/behaviours/well-being that occurred due to the pandemic (COVID-19-related stressors) and, *background characteristics* that might act as risk or protective factors. We also include *proxy baseline information* (emotional problems and SWB measured 10 years earlier). The characteristics of the sample population are shown in online supplemental table 5. Online supplemental figure 1 gives an overview of the variables used in the analysis and the respective ages when they were measured.

The first set of *COVID-19-related stressors* include perceived COVID-19 infection risk, the extent to which people practise self-isolation (having left the house in the past 7 days), increased household responsibilities (including spending more time caring for children, on household chores or working in a family business), suffering from any adverse economic events (including increases in the price of food, incurring increased health expenditures, fewer clients in a family business, and if so, whether the household reduced food consumption to cope with it) and changes in working status compared with before the pandemic. In further analysis, for the 19-year-old cohort, we replaced working status with engagement with education, given that more than half of this group were still enrolled when the pandemic began (online supplemental tables 6 and 7). Finally, among the *COVID-19-related stressors*, we included the change in SWB

**Table 1** Rates of at least mild depression and anxiety in Ethiopia, India, Peru and Vietnam

| | Ethiopia | | India | | Peru | | Vietnam | |
|---|---|---|---|---|---|---|---|---|
| | % at least mild anxiety | 95% CI | % at least mild anxiety | 95% CI | % at least mild anxiety | 95% CI | % at least mild anxiety | 95% CI |
| **Rates of at least mild anxiety** | | | | | | | | |
| **Total** | **17.87** | **16.28 to 19.54** | **11.06** | **9.88 to 12.32** | **40.86** | **38.63 to 43.12** | **9.32** | **8.16 to 10.58** |
| Male | 17.48 | 15.30 to 19.65 | 9.28*** | 7.74 to 10.81 | 33.65*** | 30.65 to 36.64 | 7.92** | 6.33 to 9.51 |
| Female | 18.32 | 15.93 to 20.70 | 13.01 | 11.14 to 14.87 | 48.28 | 45.06 to 51.49 | 10.63 | 8.88 to 12.39 |
| Rural | 14.67*** | 12.65 to 16.70 | 11.81* | 10.34 to 13.27 | 33.71*** | 28.79 to 38.63 | 9.07 | 7.47 to 10.67 |
| Urban | 21.61 | 19.06 to 24.16 | 9.20 | 7.13 to 11.27 | 42.52 | 40.04 to 45 | 9.62 | 7.83 to 11.40 |
| Poorest tercile | 14.90** | 12.29 to 17.51 | 13.52*** | 11.26 to 15.78 | 36.92** | 33.01 to 40.84 | 11.67*** | 9.41 to 13.92 |
| Middle/richest terciles | 19.32 | 17.29 to 21.34 | 9.82 | 8.42 to 11.21 | 42.63 | 39.94 to 45.32 | 8.11 | 6.74 to 9.49 |
| No internet | 15.97** | 13.82 to 18.11 | 12.16 | 8.43 to 15.89 | 51.32* | 40 to 62.63 | 22.86*** | 8.74 to 36.98 |
| With internet | 19.87 | 17.47 to 22.27 | 10.92 | 9.65 to 12.19 | 40.42 | 38.16 to 42.68 | 9.11 | 7.92 to 10.30 |
| **Rates of at least mild depression** | | | | | | | | |
| **Total** | **15.44** | **13.95 to 17.02** | **9.92** | **8.80 to 11.12** | **31.58** | **29.49 to 33.74** | **9.49** | **8.33 to 10.77** |
| Male | 14.92 | 12.88 to 16.96 | 9.64 | 8.08 to 11.21 | 27.38*** | 24.55 to 30.21 | 7.65*** | 6.09 to 9.22 |
| Female | 16.04 | 13.77 to 18.31 | 10.22 | 8.54 to 11.89 | 35.91 | 32.83 to 39.00 | 11.22 | 9.42 to 13.02 |
| Rural | 12.89*** | 10.98 to 14.81 | 10.84** | 9.43 to 12.25 | 26.40** | 21.82 to 30.99 | 9.39 | 7.77 to 11.01 |
| Urban | 18.43 | 16.03 to 20.83 | 7.60 | 5.70 to 9.50 | 32.79 | 30.44 to 35.14 | 9.62 | 7.83 to 11.40 |
| Poorest tercile | 14.35 | 11.78 to 16.91 | 10.80 | 8.74 to 12.85 | 26.15*** | 22.59 to 29.72 | 8.46 | 6.51 to 10.42 |
| Middle/richest terciles | 15.97 | 14.10 to 17.85 | 9.47 | 8.10 to 10.85 | 34.02 | 31.45 to 36.60 | 10.03 | 8.51 to 11.54 |
| No internet | 14.09* | 12.06 to 16.13 | 10.14 | 6.69 to 13.58 | 31.58 | 21.05 to 42.11 | 14.29 | 2.52 to 26.05 |
| With internet | 16.85 | 14.60 to 19.11 | 9.89 | 8.67 to 11.10 | 31.58 | 29.44 to 33.73 | 9.42 | 8.22 to 10.63 |
| N | 2183 | | 2622 | | 1887 | | 2296 | |

If any missing answers to questions then the whole score is set to missing. ***P<0.01; **p<0.05; *p<0.1. Stars represent significance of t-test of equality between groups (male–female; rural–urban; bottom–top/middle wealth tercile; internet access through home computer/working smartphone (no–yes)). Poorest and middle/richest terciles refer to the household's position in the 2016 (round 5) wealth distribution. Internet access refers to having internet access through a home computer and/or a working smartphone.

between 2016 and the pandemic. SWB was measured in round 5 (2016) at ages 15 and 22 years, and in the phone survey. Cantril's Ladder[54] asks respondents to visualise a ladder of nine steps; the bottom (top) step representing their worst (best) possible life. Respondents are asked to identify which step they presently stand on. The difference in SWB is a continuous variable ranging from –8 to +8.

The second set of *background characteristics* include individual-level and household-level risk and protective factors: gender, age, location, as well as long-term health problems and past household wealth,[55] both measured in 2016, and parent–child and peer–child relationships, measured using the total raw scores of the Marsh Self-Description Questionnaires II[56] and I.[57] Both scores range between 8 and 32, with higher scores representing more positive relationships. Peer relationships were obtained at ages 15 (younger cohort) and 22 years (older cohort) in 2016. Parent relationships were obtained at ages 15 (younger cohort, 2016) and 19 years (older cohort, 2013).

GAD-7 and PHQ-8 were not measured in previous survey rounds. Therefore, we control for *proxy baseline information* including emotional problems and SWB, both available for the 25-year-old cohort only in 2009 (round 3) at the age of 15 years (online supplemental tables 8 and 9). The Emotional Problem Scale comes from the self-completed Strengths and Difficulties Questionnaire. Total scores range from 0 to 10, a higher score indicates more severe emotional problems. We report extensive disaggregated rates of mental health issues (online supplemental tables 10–15).

Changes in responsibilities, the labour market and education environment may affect men and women differently. Therefore, we re-estimated the regressions separately by gender (online supplemental tables 16–23). We report ORs, robust SEs and the 95% CIs for all regressions.

### Patient and public involvement

No patients or the public were involved in the study design, setting the research questions, interpretation or writing up of results, or reporting of the research as it is a prospective cohort study.

### RESULTS

Across the four countries, 93% of the Young Lives sample, who were located during the last tracking exercise in November 2019, participated in the phone survey. Only between 0.1% (Vietnam) and 2% (Peru) did not respond to the mental health questions. The results presented in this section refer to the participants aged 19 years and 25 years together, unless differently specified.

### Country comparisons

Both mild anxiety (41%, 95% CI 38.63% to 43.12%) and mild depression (32%, 95% CI 29.49% to 33.74%) rates were highest in Peru, followed by Ethiopia (anxiety:

18%, 95% CI 16.28% to 19.54%; depression: 15%, 95% CI 13.95% to 17.02%) (see table 1). The rates of moderate/severe anxiety and depression were highest in Peru, 13.5% (95% CI 12.00% to 15.14%) and 9.6% (95% CI 8.35% to 11.07%), and below 3% in the other countries (online supplemental tables 24–31). Women had significantly higher rates of anxiety symptoms in all countries except Ethiopia and higher rates of depression in Peru and Vietnam. In Peru, almost half of all women had symptoms consistent with at least mild anxiety. Rates of anxiety and depression in rural areas were significantly lower than urban rates in Ethiopia and Peru, but significantly higher in India. The poorest wealth tercile had significantly lower rates of anxiety in Ethiopia and Peru, but higher rates in India and Vietnam. In Peru, the poorest wealth tercile also had significantly lower rates of depression. Those not having any access to the internet, although a minority, had significantly higher levels of anxiety (Vietnam p<0.01, Peru p<0.1).

We note a high correlation between GAD-7 and PHQ-8 scores (minimum 0.610 (p<0.01) (India) and maximum 0.700 (p<0.01) (Peru)), and the rate of having both (at least mild) anxiety and depression symptoms was high, with values of up to 24.8% (95% CI 22.87% to 26.81%) in Peru (online supplemental tables 32 and 33). We use Pearson's correlation coefficient with Bonferroni corrected p values to investigate (1) the relationship between GAD-7 and PHQ-8 raw scores and (2) the relationship between SWB and GAD-7 and PHQ-8 raw scores.

The significant risk and protective factors were similar. For brevity, the main results refer to associations with experiencing at least mild anxiety symptoms (see table 2), pooling the two cohorts and women and men together (unless differently specified). At the end, we comment on the differences between these results and those for depression (see table 3).

### Logistic regression results (ORs): at least mild anxiety
#### COVID-19-related stressors

*COVID-19 infection risk perception:* The odds of those who believed that they were at medium/high risk of catching the virus were 1.27 (95% CI 0.98 to 1.63, p<0.1) (India) to 1.46 (95% CI 1.20 to 1.78, p<0.01) (Peru) times higher than for those who believed themselves to have no/low risk. The former group had rates of at least mild anxiety of 12% (India) and 45% (Peru).

*Leaving the house for at least 1 day a week*: No significant effects, except in India where it increased the odds of anxiety by 1.40 (95% CI 0.96 to 2.06, p<0.1).

*Economic adversity*: For those who suffered from economic adversity (eg, fewer clients in the family business, food price increase), odds of anxiety were higher (p<0.01 (Ethiopia and Vietnam), (2.50, 95% CI 1.14 to 5.46, p<0.05) (2.40, 95% CI 1.01 to 5.72, p<0.05) (Peru)) even if it did not cause reduced food consumption. Moreover, in Ethiopia and Vietnam, those who reduced food consumption as a coping strategy had 7.19 (95% CI 4.51 to 11.45, p<0.01) (Ethiopia) and 1.67 (95% CI

**Table 2** Logistic regression results: symptoms of at least mild anxiety

| | Ethiopia | | India | | Peru | | Vietnam | |
|---|---|---|---|---|---|---|---|---|
| | OR | 95%CI | OR | 95%CI | OR | 95%CI | OR | 95%CI |
| COVID-19-related stressors | | | | | | | | |
| Risk perception: believe they are at medium/high risk | 0.813 (0.11) | 0.629 to 1.052 | 1.265* (0.16) | 0.980 to 1.633 | 1.460*** (0.15) | 1.200 to 1.777 | 1.022 (0.19) | 0.717 to 1.459 |
| Left house for any reason in the past 7 days | 0.817 (0.20) | 0.501 to 1.332 | 1.401* (0.27) | 0.955 to 2.056 | 0.994 (0.13) | 0.776 to 1.274 | 1.054 (0.25) | 0.666 to 1.668 |
| Difference in subjective well-being R5–Call2 | 1.070** (0.03) | 1.012 to 1.133 | 1.099** (0.04) | 1.020 to 1.184 | 1.105*** (0.03) | 1.053 to 1.159 | 1.181*** (0.05) | 1.081 to 1.291 |
| Change in responsibilities | | | | | | | | |
| Spend more time taking care of children | 0.891 (0.14) | 0.652 to 1.219 | 2.209*** (0.37) | 1.593 to 3.063 | 1.354*** (0.14) | 1.100 to 1.667 | 1.401* (0.25) | 0.986 to 1.991 |
| Spend more time on household chores | 1.059 (0.16) | 0.794 to 1.414 | 0.756* (0.11) | 0.565 to 1.011 | 1.204 (0.15) | 0.943 to 1.537 | 1.209 (0.19) | 0.888 to 1.645 |
| Spend more time working in the family business | 0.772 (0.16) | 0.520 to 1.149 | 1.605* (0.46) | 0.916 to 2.811 | 1.288* (0.18) | 0.976 to 1.699 | 1.790*** (0.37) | 1.196 to 2.679 |
| Economic adversities | | | | | | | | |
| Faced with new health expenses | 0.937 (0.18) | 0.646 to 1.359 | 1.046 (0.14) | 0.798 to 1.370 | 1.733*** (0.19) | 1.404 to 2.139 | 1.240 (0.33) | 0.733 to 2.098 |
| Experienced adversity but did not reduce food consumption | 2.362*** (0.49) | 1.567 to 3.561 | 0.621 (0.28) | 0.253 to 1.524 | 2.495*** (1.00) | 1.140 to 5.461 | 1.624*** (0.2) | 1.161 to 2.270 |
| Reduced food consumption as response to experienced adversity | 7.187*** (1.71) | 4.510 to 11.454 | 0.871 (0.46) | 0.310 to 2.446 | 2.402** (1.06) | 1.009 to 5.718 | 1.673** (0.37) | 1.089 to 2.570 |
| Changes in employment status | | | | | | | | |
| Did not work before the pandemic, but is working now | 2.667*** (0.64) | 1.671 to 4.259 | 0.993 (0.22) | 0.643 to 1.533 | 0.866 (0.16) | 0.609 to 1.232 | 1.104 (0.35) | 0.594 to 2.053 |
| Worked before the pandemic and is working now/has a job | 1.570*** (0.24) | 1.156 to 2.132 | 1.288 (0.22) | 0.925 to 1.795 | 1.068 (0.15) | 0.815 to 1.399 | 0.978 (0.20) | 0.653 to 1.464 |
| Worked before the pandemic and is not working now/does not have a job | 2.294*** (0.45) | 1.559 to 3.376 | 2.504*** (0.65) | 1.501 to 4.175 | 1.205 (0.18) | 0.893 to 1.628 | 1.346 (0.33) | 0.834 to 2.171 |
| Educational disruption | | | | | | | | |

**Table 2** Continued

| | Ethiopia | | India | | Peru | | Vietnam | |
|---|---|---|---|---|---|---|---|---|
| | OR | 95% CI | OR | 95% CI | OR | 95% CI | OR | 95% CI |
| Enrolled in/planning to enrol in full-time education and not participating in learning activities† | 1.588** (0.33) | 1.052 to 2.396 | 1.043 (0.21) | 0.707 to 1.538 | 0.600 (0.27) | 0.245 to 1.471 | 0.828 (0.25) | 0.458 to 1.495 |
| Enrolled in/planning to enrol in full-time education and participating in learning activities† | 1.110 (0.38) | 0.565 to 2.182 | 0.710 (0.17) | 0.450 to 1.120 | 1.026 (0.12) | 0.816 to 1.290 | 0.699* (0.15) | 0.465 to 1.051 |
| Background characteristics | | | | | | | | |
| Age in months | 1.008*** (0.00) | 1.005 to 1.011 | 1.001 (0.00) | 0.998 to 1.005 | 0.999 (0.00) | 0.997 to 1.002 | 0.995** (0.00) | 0.991 to 0.999 |
| Female | 1.074 (0.16) | 0.810 to 1.426 | 1.601*** (0.27) | 1.149 to 2.231 | 1.706*** (0.18) | 1.389 to 2.095 | 1.295* (0.20) | 0.961 to 1.746 |
| Urban | 1.276* (0.18) | 0.974 to 1.670 | 0.903 (0.15) | 0.658 to 1.238 | 1.441** (0.22) | 1.072 to 1.937 | 1.206 (0.18) | (0.895 to 1.625) |
| Participant has long-term health problem, 2016 (round 5) | 1.424* (0.27) | 0.984 to 2.060 | 1.399* (0.25) | 0.989 to 1.981 | 1.805*** (0.26) | 1.354 to 2.406 | 1.337 (0.27) | 0.897 to 1.991 |
| Total parent–child relationship score, 2012/2016 (round 4/5) | 0.967 (0.02) | 0.926 to 1.009 | 0.923*** (0.02) | 0.883 to 0.965 | 0.976* (0.01) | 0.949 to 1.004 | 1.041 (0.03) | 0.992 to 1.093 |
| Total peer–child relationship score, 2016 (round 5) | 0.941*** (0.02) | 0.901 to 0.982 | 1.025 (0.03) | 0.975 to 1.078 | 1.001 (0.02) | 0.970 to 1.033 | 0.941* (0.03) | 0.876 to 1.011 |
| Middle/top wealth tercile R5, 2016 (round 5) | 1.124 (0.16) | 0.846 to 1.495 | 0.781* (0.11) | 0.587 to 1.038 | 1.016 (0.13) | 0.797 to 1.295 | 0.550*** (0.09) | 0.404 to 0.747 |
| Proxy baseline information | | | | | | | | |
| Emotional Problem Scale score, 2009 (round 3)‡ | 1.074* (0.04) | 0.994 to 1.161 | 1.007 (0.05) | 0.910 to 1.114 | 1.222*** (0.07) | 1.093 to 1.366 | 1.072 (0.07) | 0.940 to 1.221 |
| Subjective well-being, 2009 (round 3)‡ | 0.979 (0.06) | 0.872 to 1.100 | 1.131** (0.07) | 1.010 to 1.267 | 0.924 (0.07) | 0.800 to 1.068 | 0.975 (0.08) | 0.827 to 1.148 |
| N | 2183 | | 2622 | | 1887 | | 2296 | |

ORs are unadjusted ORs. Robust SEs in parenthesis, ***significant at 1%, **significant at 5%, *significant at 10%. Base categories are as follows: believe they are at no/low risk, did not leave the house at all during the past 7 days, did not spend more time taking care of children, did not spend more time working in the family business, did not face new health expenses, did not suffer a shock, did not work at all in the past 12 months OR worked during the pandemic but not before and is not working now, never attended school or not enrolled in full-time education/not planning to enrol, male, rural, does not have a long-term health condition, lowest wealth tercile. All time-variant variables are measured in 2020 unless otherwise specified. The regression was run on the joint younger cohort /older cohort sample except for the results which were added from independent regression specifications run on the younger and older cohort only.

†Results come from a separate regression where educational disruption was substituted for changes in employment status and which was run on the younger cohort (18–19 years) sample only (online supplemental table 6).

‡Results come from a separate regression where we added proxy baseline information measured in 2009 at age 15 years and which was run on the older cohort (25–26 years) sample only (online supplemental table 8).

**Table 3** Logistic regression results: symptoms of at least mild depression

| | Ethiopia | | India | | Peru | | Vietnam | |
|---|---|---|---|---|---|---|---|---|
| | OR | 95% CI | OR | 95% CI | OR | 95% CI | OR | 95% CI |
| **COVID-19-related stressors** | | | | | | | | |
| Risk perception: believe they are at medium/high risk | 0.620*** (0.08) | 0.481 to 0.800 | 0.962 (0.13) | 0.736 to 1.258 | 1.595*** (0.17) | 1.297 to 1.963 | 1.107 (0.20) | 0.778 to 1.575 |
| Left house for any reason in the past 7 days | 0.995 (0.26) | 0.596 to 1.661 | 2.285*** (0.53) | 1.454 to 3.590 | 1.055 (0.14) | 0.813 to 1.368 | 0.965 (0.21) | 0.629 to 1.480 |
| Difference in subjective well-being R5-Call2 | 1.089*** (0.03) | 1.026 to 1.155 | 1.119*** (0.04) | 1.041 to 1.202 | 1.080*** (0.03) | 1.026 to 1.137 | 1.163*** (0.05) | 1.068 to 1.267 |
| **Change in responsibilities:** | | | | | | | | |
| Spend more time taking care of children | 0.852 (0.14) | 0.623 to 1.165 | 1.611** (0.30) | 1.121 to 2.316 | 1.145 (0.13) | 0.919 to 1.425 | 1.455** (0.26) | 1.021 to 2.074 |
| Spend more time on household chores | 1.128 (0.17) | 0.834 to 1.526 | 0.877 (0.14) | 0.649 to 1.186 | 1.061 (0.14) | 0.821 to 1.371 | 1.550*** (0.25) | 1.133 to 2.119 |
| Spend more time working in the family business | 0.948 (0.20) | 0.633 to 1.419 | 1.053 (0.35) | 0.551 to 2.011 | 1.302* (0.19) | 0.975 to 1.738 | 1.804*** (0.35) | 1.228 to 2.650 |
| **Economic adversities** | | | | | | | | |
| Faced with new health expenses | 0.522*** (0.11) | 0.342 to 0.796 | 1.289* (0.19) | 0.967 to 1.716 | 1.739*** (0.19) | 1.401 to 2.159 | 0.909 (0.26) | 0.523 to 1.580 |
| Experienced adversity but did not reduce food consumption | 3.808*** (0.95) | 2.336 to 6.206 | 0.646 (0.32) | 0.244 to 1.713 | 2.049* (0.86) | 0.896 to 4.686 | 1.695*** (0.30) | 1.203 to 2.390 |
| Reduced food consumption as response to experienced adversity | 10.890*** (3.00) | 6.341 to 18.701 | 0.641 (0.38) | 0.203 to 2.027 | 2.531** (1.17) | 1.019 to 6.285 | 1.907*** (0.41) | 1.249 to 2.912 |
| **Changes in employment status** | | | | | | | | |
| Did not work before the pandemic, but is working now | 2.657*** (0.64) | 1.655 to 4.265 | 0.685* (0.16) | 0.438 to 1.072 | 0.920 (0.17) | 0.643 to 1.316 | 0.715 (0.24) | 0.375 to 1.363 |
| Worked before the pandemic and is working now/has a job | 1.378** (0.22) | 1.010 to 1.881 | 0.873 (0.15) | 0.620 to 1.229 | 0.841 (0.12) | 0.634 to 1.114 | 0.810 (0.16) | 0.552 to 1.190 |
| Worked before the pandemic and is not working now/does not have a job | 1.679** (0.36) | 1.104 to 2.555 | 1.597 (0.45) | 0.914 to 2.790 | 1.283 (0.20) | 0.940 to 1.751 | 1.509* (0.34) | 0.974 to 2.336 |
| **Educational disruption** | | | | | | | | |
| Enrolled in/planning to enrol in full-time education and not participating in learning activities† | 1.728*** (0.36) | 1.153 to 2.592 | 1.103 (0.22) | 0.743 to 1.638 | 0.743 (0.36) | 0.287 to 1.928 | 0.944 (0.28) | 0.529 to 1.685 |

Continued

**Table 3** Continued

| | Ethiopia | | India | | Peru | | Vietnam | |
|---|---|---|---|---|---|---|---|---|
| | OR | 95% CI | OR | 95% CI | OR | 95% CI | OR | 95% CI |
| Enrolled in/planning to enrol in full-time education and participating in learning activities† | 1.174 (0.41) | 0.591 to 2.333 | 0.749 (0.18) | 0.469 to 1.196 | 1.042 (0.13) | 0.816 to 1.329 | 1.172 (0.25) | 0.768 to 1.788 |
| Background characteristics | | | | | | | | |
| Age in months | 1.001 (0.00) | 0.998 to 1.005 | 1.000 (0.00) | 0.997 to 1.004 | 0.996** (0.00) | 0.993 to 0.999 | 0.994*** (0.00) | 0.989 to 0.998 |
| Female | 1.142 (0.17) | 0.847 to 1.541 | 1.239 (0.21) | 0.890 to 1.725 | 1.317** (0.15) | 1.059 to 1.637 | 1.405** (0.21) | 1.048 to 1.883 |
| Urban | 1.356** (0.20) | 1.019 to 1.805 | 0.733* (0.12) | 0.526 to 1.022 | 1.114 (0.17) | 0.820 to 1.513 | 0.961 (0.15) | 0.706 to 1.308 |
| Participant has long-term health problem, 2016 (round 5) | 0.954 (0.21) | 0.620 to 1.470 | 1.566** (0.28) | 1.101 to 2.226 | 1.590*** (0.23) | 1.197 to 2.113 | 1.328 (0.27) | 0.889 to 1.982 |
| Total parent–child relationship score, 2012/2016 (round 4/5) | 1.004 (0.02) | 0.960 to 1.050 | 0.909*** (0.02) | 0.867 to 0.954 | 0.939*** (0.01) | 0.911 to 0.968 | 0.983 (0.02) | 0.938 to 1.031 |
| Total peer–child relationship score, 2016 (round 5) | 0.938*** (0.02) | 0.895 to 0.983 | 1.015 (0.03) | 0.963 to 1.070 | 1.007 (0.02) | 0.973 to 1.042 | 0.988 (0.04) | 0.920 to 1.062 |
| Middle/top wealth tercile R5, 2016 (round 5) | 0.884 (0.13) | 0.656 to 1.192 | 1.074 (0.16) | 0.802 to 1.437 | 1.249* (0.16) | 0.968 to 1.613 | 1.068 (0.18) | 0.770 to 1.482 |
| Proxy baseline information | | | | | | | | |
| Emotional Problem Scale score, 2009 (round 3)‡ | 1.090* (0.05) | 0.995 to 1.195 | 1.101* (0.06) | 0.991 to 1.223 | 1.129** (0.07) | 1.003 to 1.270 | 1.114 (0.08) | 0.973 to 1.276 |
| Subjective well-being, 2009 (round 3)‡ | 0.894 (0.06) | 0.782 to 1.023 | 1.077 (0.07) | 0.952 to 1.219 | 0.869* (0.07) | 0.736 to 1.026 | 0.952 (0.08) | 0.805 to 1.126 |
| N | 2183 | | 2622 | | 1887 | | 2296 | |

ORs are unadjusted ORs. Robust SEs in parenthesis, ***significant at 1%, **significant at 5%, *significant at 10%. Base categories are as follows: believe they are at no/low risk, did not leave the house at all during the past 7 days, did not spend more time taking care of children, did not spend more time on household chores, did not spend more time working in the family business, did not face new health expenses, did not suffer a shock, did not work at all in the past 12 months OR worked during the pandemic but not before and is not working now, never attended school or not enrolled in full-time education/not planning to enrol, male, rural, does not have a long-term health condition, lowest wealth tercile. All time-variant variables are measured in 2020 unless otherwise specified. The regression was run on the joint younger cohort /older cohort sample except for the results which were added from independent regression specifications run on the younger and older cohort only.

†Results come from a separate regression where educational disruption was substituted for changes in employment status and which was run on the younger cohort (19–20 years) sample only (online supplemental table 7).

‡Results come from a separate regression where we added proxy baseline information measured in 2009 at age 15 years and which was run on the older cohort (25–26 years) sample only (online supplemental table 9).

1.09 to 2.57, p<0.05) (Vietnam) higher odds than those who experienced an adverse event but did not need to reduce food consumption in response 2.36 (95% CI 1.57 to 3.56, p<0.01) (Ethiopia) and 1.62 (95% CI 1.16 to 2.27, p<0.01) (Vietnam) (both compared with those who did not experience any adverse event at all). In Ethiopia, 36% of those who reduced food consumption reported at least mild anxiety compared with 7% of those who did not experience an adverse event (p<0.0001). In Ethiopia, odds were higher among women than men, but in Peru and Vietnam significant for men only. Facing new health expenses significantly increased the odds by 1.73 (95% CI 1.40 to 2.14) (p<0.01) in Peru. Over half (52%, p<0.0001) of those who faced new health expenses report at least mild anxiety (although not significant in India or Ethiopia, significant risk factor for women in Vietnam (1.76, 95% CI 0.93 to 3.31, p<0.1)).

### Increased responsibilities
Spending more time on childcare during the lockdown increased odds of anxiety by 2.21 (95% CI 1.59 to 3.06, p<0.01) in India, 1.35 (95% CI 1.10 to 1.67, p<0.01) in Peru and 1.4 (95% CI 0.99 to 1.99, p<0.1) in Vietnam. Rates for those who spent more time taking care of children were 20% vs 9% (India), 49% vs 37% (Peru) and 13% vs 8% (Vietnam), (all p<0.001). Spending more time on household chores *lowered* odds in India for anxiety (women only). For those who spend more time working in the family business, the odds of anxiety were 1.61 (95% CI 0.92 to 2.81, p<0.1) times higher in India, 1.29 (95% CI 0.98 to 1.67, p<0.1, n.s. for women) in Peru and 1.80 (95% CI 1.20 to 2.68, p<0.01) in Vietnam (higher odds among men). In Vietnam, those who spent more time working in the family business reported rates of 16% (p<0.0001), the highest among the Vietnamese sample.

### Changes in employment status
In Ethiopia, those who participated in the labour market had higher odds of anxiety than those who did not (eg, full-time students, stay-at-home parents). However, the odds of those who were pushed into the labour market (2.67, 95% CI 1.67 to 4.26, p<0.01) or lost their job (2.29, 95% CI 1.56 to 3.38, p<0.01) were higher than those who simply participated (1.57, 95% CI 1.16 to 2.13, p<0.01) (all in comparison with non-participants). In India, losing a job increased the risk of anxiety by 2.50 (95% CI 1.50 to 4.18, p<0.01). In Peru and Vietnam, there were no employment effects. Rates of at least mild anxiety among those who lost their jobs were among the highest in each country 31% (Ethiopia, p<0.001), 20% (India, p<0.001), 46% (Peru, n.s.), 12% (Vietnam, n.s.).

### Educational disruption (19-year-old cohort only)
Students who were enrolled in Ethiopia before the pandemic and were unable to access virtual classes or complete homework had 1.59 (95% CI 1.05 to 2.40, p<0.05) times higher odds of anxiety than those who were not enrolled. In Vietnam, those who were enrolled and engaged in learning activities had lower odds of anxiety (0.70, 95% CI 0.47 to 1.05, p<0.1) than those who were not enrolled (base category). The full younger cohort only regression results can be found in online supplemental table 6; the education results split by gender (again younger cohort only) are located in online supplemental tables 20–23.

### Background characteristics
For *women*, the odds were 1.30 (95% CI 0.96 to 1.75, p<0.1) (Vietnam), 1.60 (95% CI 1.15 to 2.23, p<0.01) (India) and 1.70 (95% CI 1.39 to 2.10, p<0.01) (Peru) times greater than the odds for men (n.s. in Ethiopia). Urban location increased odds significantly in Ethiopia and Peru. Age was not significant in India and Peru, protective in Vietnam and a risk factor in Ethiopia.

*Long-term health problems* (measured in 2016): the odds of at least mild anxiety were 1.42 (95% CI 0.98 to 2.06, p<0.1) (Ethiopia), 1.40 (95% CI 1.00 to 2.00, p<0.1) (India) and 1.80 (95% CI 1.35 to 2.41, p<0.01) (Peru) times as large as the odds for those who did not (n.s. in Vietnam). In Peru, those reporting long-term health problems had the highest rates of at least mild anxiety, 56% (p<0.0001).

*Parent–child relationship* (measured at age 15 years for the younger cohort, in 2016 and measured at age 19 years for the older cohort, in 2013) and *peer–child relationship* (at age 15 years for the younger cohort and at age 22 years for the older cohort, in 2016): strong parent–child relationships were a significant protective factor in India and Peru, while peer–child relationships were a significant protective factor in Ethiopia and Vietnam.

*Past household wealth* (measured in 2016): that is, being in the middle/highest wealth tercile versus the lowest was a marginally significant protective factor in India (0.78, 95% CI 0.59 to 1.04, p<0.1) and significant in Vietnam (0.55, 95% CI 0.40 to 0.75, p<0.01).

### Proxy baseline information
*Past emotional problems and well-being* (25-year-old cohort only): for a one-point increase in previous emotional problems at age 15 years (measured in 2009), the odds of at least mild anxiety increased by a factor of 1.22 (95% CI 1.09 to 1.37, p<0.01) (Peru) and 1.07 (95% CI 0.99 to 1.16, p<0.1) (Ethiopia). Notably, the effect of the COVID-19-related stressors holds when controlling for past proxy baseline information. The full older cohort only regression results can be found in online supplemental table 8.

### Significant differences between anxiety and depression logistic regression results
As previously mentioned, the results for anxiety and depression (see table 3 and online supplemental tables 7, 9, 16–23, 34–36) are qualitatively similar. Here we note significant differences in the results between the two independent variables for each country. In Ethiopia, food insecurity had a higher impact on depression for men than women. In India, subjective high infection risk

increased anxiety, but not depression, and those faced with new health expenses had higher odds of depression, but not anxiety. Women also had higher rates of anxiety but not depression. In Peru, childcare was not a risk factor for depression, and past SWB was a protective factor. In Vietnam, losing a job was a significant risk factor for depression, while good peer relations and education were not significant determinants of depression.

## DISCUSSION

We examined the impact of the COVID-19 pandemic on the mental health of young people in Ethiopia, India, Peru and Vietnam. The sample has broad coverage of the poorer population in each country, and we interviewed 93% of those located during the tracking prior to the pandemic, including those without internet access and in Ethiopia also those without mobile phone, who would be excluded from an online survey. Internet access has had both positive and negative effects in the pandemic.[58] In our sample, those without access to internet have significantly higher rates of anxiety in Vietnam and Peru.

The four countries have had different experiences of the pandemic—Peru is the most affected country in terms of deaths per population, and Vietnam the least. While Vietnam has recorded only 35 deaths in total (not per day), with no deaths since 3 September 2020, on 15 October (end date of our survey) Peru had registered 33 577 deaths.[59] Another aspect which has likely contributed to the difference is the length of the lockdown, which creates stress and reduces household income. In Peru, this was 107 consecutive days at the national level, followed by an additional period of local lockdowns, such that certain areas of the country were in lockdown for up to 199 days. In contrast, Vietnam had a very short and successful lockdown, one further localised lockdown, but by September life was already back to normal. Ethiopia restricted certain activities and closed schools, but did not impose a strict lockdown, though faced other challenges (locust infestations, food price inflation and violence).

This study reveals a strong relationship between the severity of the pandemic and the rates of mental health conditions in our sample, both in terms of anxiety and depression symptoms. Rates of at least mild anxiety (depression) were four (three) times higher in Peru compared with Vietnam. Furthermore, the 2020 survey showed a significant fall in SWB from 2016 in all countries except Vietnam. The fall in SWB is highly correlated with anxiety and depression symptoms. In the absence of baseline measures of GAD-7 and PHQ-8, a strong correlation between SWB and our mental health indicators is important, as it suggests that SWB is a useful proxy baseline.

The economic impact of the pandemic has affected certain groups of young people in all study countries, even Vietnam, and Ethiopia where there was no full national lockdown. Overall, our findings confirmed that those experiencing COVID-19-related stressors had worse mental health, although the relative importance of stressors varied across countries: increased health expenses and believing they were at a medium/high infection risk was detrimental for young people in Peru, but increased food insecurity was much more important in Ethiopia, reflecting high rates of food price inflation in 2019, which continued into 2020. Moreover, good peer relations in earlier years were a protective factor for anxiety and depression only in Ethiopia. In Peru and Vietnam, there were no employment effects on anxiety, likely for very different reasons—in Peru health concerns were more important, and in Vietnam, the labour market was relatively resilient.

Exploiting the longitudinal data allowed us to investigate individual-level and household-level protective and risk factors. As expected, parent and peer relations measured during childhood and adolescence were protective, though in different ways across countries. Strong parental relationships were a significant protective factor in India and Peru, whereas peer relationships were more important in Ethiopia and Vietnam. Those reporting long-term health problems were twice as likely to display symptoms consistent with at least mild anxiety, this effect being particularly pronounced in Peru. Previous relative wealth was a significant protective factor only in India and Vietnam. Pre-pandemic emotional problems were risk factors, especially in Ethiopia and Peru. The associations with COVID-19-related stressors were robust to the inclusion of pre-pandemic emotional problems and past SWB.

Other studies have used longitudinal data to document the impact of the pandemic on mental health,[5 12] though none investigate a comparable population of young people, of a similar age, with those in our study countries, though results from the UK have similar findings. The closest study to ours, a phone survey in a developing country, finds a deterioration in maternal mental health in rural Bangladesh.[12] Our study shows lower rates of anxiety and depression in rural areas in Ethiopia and Peru, but significantly higher in India. We are able to disaggregate the effect of a range of COVID-19-related stressors, which we can relate individually to other studies. A study in Hubei province, China[11] showed the importance of income losses during the pandemic. Studies of college students in China[8] and Bangladesh[10] show that educational disruption significantly increased anxiety and depression, similar to our results in Vietnam and Ethiopia. Social support was negatively correlated with the level of anxiety,[8] similar to our findings regarding parent/peer–child relationships. In Jordan,[9] female healthcare professionals, female university students and university students with chronic disease were at higher risk of developing depression, similar to our results for long-term health problems.

Even controlling for other factors, we found women to be more vulnerable to anxiety in India, Peru and Vietnam and more vulnerable to depression in Peru and Vietnam. which is similar to most COVID-19-related[5 9 11] and pre-pandemic studies.[60 61] However, in Ethiopia, we found no

significant gender effects. Previous studies in Ethiopia had mixed results on gender differences.[62 63] Relatedly, and during the COVID-19 pandemic, one[8] study finds no gender differences among Chinese college students while another[10] finds that male Bangladeshi students had higher depressive symptoms than women.

## Strengths and limitations

This study's strength combines survey data about experiences of COVID-19 with long-term information from two cohorts of participants of a population-based cohort study in four LMICs. The study was able to cover the poorest, those without internet (and without mobile phone in Ethiopia), and examine the role of a broad range of pandemic-related stressors and of individual-level and household-level risk and protective factors. Our study has a number of limitations. We do not have a directly comparable pre-COVID-19 baseline for depression/anxiety. However, we use proxy variables for baseline and our explanatory variables capture dynamics during the pandemic. As for other studies, there may be under-reporting in all four countries[64–67] because of stigma associated with mental health, despite piloting and validation. There is evidence in the literature of stigmatisation of mental health in all four countries, as in other LMIC meta-analyses,[68 69] but nothing to suggest this is associated with the differences we find. Furthermore, our analysis identifies high-risk groups *within* each country. Additionally, self-reported variables may be biased due to feelings of anxiety or depression. The findings are not fully generalisable to the whole population of LMICs due to the poverty-focused design and age group, however they broadly represent poor young people in the study countries.

## CONCLUSION

Adolescents and young people have been a lower priority for COVID-19 interventions, given the lower rates of hospitalisation and death for this age group. This research shows that the pandemic is having important effects on the mental health of certain groups of young people, even in countries with fewer cases. Mental health services are very limited in LMICs, making it urgent to develop evidence-based and sustainable prevention programmes in response to the pandemic. As a short-term measure, funding for (and awareness of) telephone helplines can be increased, and Cash Transfer Programmes should be expanded or conditionality waived to cover young people hardest hit by the pandemic, and include mobile phone messaging to provide accurate non-stigmatised information about COVID-19 and available mental health support services. This could help break the cycle between poverty and mental illness, lowering the risk of long-term consequences.[70] Further research on mental health in Peru (the country hardest hit by the pandemic among our four countries) should be conducted including estimating the impact on mental health resulting from lockdown length/intensity and COVID-19 cases/deaths on the district level.

**Author affiliations**
[1]Management School, Lancaster University, Lancaster, UK
[2]Oxford Department of International Development, University of Oxford, Oxford, UK
[3]Niños del Milenio, Grupo de Análisis para el Desarrollo (GRADE), Lima, Peru
[4]Director, Centre for Economic and Social Studies, Begumpet, India
[5]Department of Economics, Addis Ababa University, Addis Ababa, Ethiopia
[6]Centre for Analysis and Forecasting, Vietnam Academy of Social Sciences, Hanoi, Vietnam
[7]Department of Psychology, University of California Los Angeles, Los Angeles, California, USA
[8]Department of Psychiatry, University of Oxford, Oxford, UK
[9]School of Public Health, University of the Witwatersrand, Johannesburg-Braamfontein, South Africa

**Acknowledgements** We particularly wish to thank the Young Lives respondents and their families for generously giving us their time and cooperation during this difficult time as well as fieldworkers and field managers in the four study countries. We are gratefully indebted to Professor George Patton for his helpful comments. Further, we would like to thank Professor Graham Thornicroft, Dr Julio Torales and Dr Gabriel Teck for their helpful feedback during the review process.

**Contributors** CP and MF conceived the study. CP, MF, DS and ASJ designed the study. MF, ASJ, RE, TW and LTD led data collection. CP and AH did the statistical analyses. CP, AH and MF wrote the first draft of the article. AS and MC provided comments and input to the several drafts of the article. MF, DS and ASJ verified the underlying data. AS, MF, CP and AH assessed scale validation and methodology. AH prepared the supplemental files. All authors critically reviewed multiple versions of the manuscript and approved the final version.

**Funding** Young Lives at Work is funded by a UK aid from the Foreign, Commonwealth and Development Office (FCDO), under the Department for International Development, UK Government (grant number 200 425).

**Disclaimer** The funders of the study had no role in study design, data collection, data analysis, data interpretation or writing of the report. The views expressed are those of the authors. They are not necessarily those of, or endorsed by, the University of Oxford, Young Lives, FCDO. All authors had full access to the anonymised data in the study and had final responsibility for the decision to submit for publication.

**Competing interests** CP, MF, AH, DS, ASJ, RE, TW and LTD report grants from the FCDO, during the conduct of the study.

**Patient consent for publication** Not required.

**Ethics approval** The survey was approved by the institutional research ethics committees at the University of Oxford (UK), the University of Addis Ababa (Ethiopia), the Centre for Economic and Social Studies in Hyderabad (India), the Instituto de Investigación Nutricional (Peru) and the Hanoi University of Public Health (Vietnam). Participants were asked for their verbal informed consent before the study commenced and were assured of confidentiality. A consultation guide was provided to all participants with resources for support in issues raised by the questionnaire, including mental health. Approval number (Oxford): CUREC 1A/ ODID CIA-20-034.

**Provenance and peer review** Not commissioned; externally peer reviewed.

**Data availability statement** Data are available in a public, open access repository. The entire individual participant data collected during the phone survey and previous in-person rounds, after de-identification, are available including data dictionaries. Furthermore, the questionnaire, attrition reports and the fieldwork manual are available at https://www.younglives.org.uk/. The data are available from January 2021, with no end date to anyone who wishes to access the data for any purpose, via the UK Data Archive (study number 8678, DOI: 10.5255/UKDA-SN-8678-1).

of the translations (including but not limited to local regulations, clinical guidelines, terminology, drug names and drug dosages), and is not responsible for any error and/or omissions arising from translation and adaptation or otherwise.

**ORCID iD**
Catherine Porter http://orcid.org/0000-0002-8578-1744

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
