## [Reviewer comments · BMJ Open]

ARTICLE DETAILS

TITLE (PROVISIONAL)	Impact of the COVID-19 Pandemic on anxiety and depression symptoms of young people in the Global South: evidence from a four-country cohort study
AUTHORS	Porter, Catherine; Favara, Marta; Hittmeyer, Annina; Scott, Douglas; Sánchez Jiménez, Alan; Ellanki, Revathi; Woldehanna, T; Duc, Le Thuc; Craske, Michelle; Stein, Alan

VERSION 1 – REVIEW

REVIEWER	Thornicroft, Graham institute of psychiatry, health service and population research
REVIEW RETURNED	08-Feb-2021

GENERAL COMMENTS	Review of: Impact of the COVID-19 Pandemic on anxiety and depression symptoms of young people in the Global South: evidence from a four-country cohort study The strengths of this paper include • Two long established cohorts of young people• A sensible array of risk and protective factors were assessed• Standardised scales were used to assess anxiety and depression• Important data are presented here on prevalence rates, and their differences across countries The paper could be improved by • Clarify how people ‘without phones or internet access’ took part in a phone survey• Saying more about why no pre baseline data on anxiety or depression were available• Discussing more about how far stigma related reluctance to disclose mental illness may at least in part account for the country difference in reported rates of anxiety and depression• Saying why there was no patient or public involvement• Tidying up errors for reference insertion points eg on page 8 etc• I think that more needs to be said about the huge differences in prevalence rates eg between Peru and Vietnam, how can these be understood? Are differences in covid related mortality rates sufficient to explain the magnitude of these differences?• What may explain why countries do or do not show gender differences in these prevalence rates?• There appear to be 36 supplementary tables, are all these necessary?
--

REVIEWER	Torales, Julio National University of Asuncion, Psychiatry
-----------------	---

REVIEW RETURNED	20-Feb-2021
GENERAL COMMENTS	Very fine piece of work. I recommend that the authors add references that support the usefulness of using telephone surveys compared to "in-person" interviews.
REVIEWER	Teck, Gabriel Universiti Teknologi Malaysia
REVIEW RETURNED	20-Feb-2021

GENERAL COMMENTS	Generally, this paper is of great importance and can be considered for publication after some revisions are performed. English is good and is grammatically sound except for some typos (please refer to the commented file- see the abstract some words are missing). Too many supplementary tables, although there are necessary. Perhaps reducing some of them or some can be embedded in the main texts. Abstract: it is well summarised but it can be improved by adding in some important points, especially for the conclusion part. Introduction: good problem state ment and research gap is well justified. But why only 4 countries and why those countries were selected? Perhaps, prior to methods, the literature review or introduction should highlight a theoretical framework providing clearer assumptions or directions of those factors/variables effects towards anxiety and depression levels. Some references can be added (please see the file attached). Methods: Why PHQ8 ? why not Phq-9? some rationales should be provided as to why GAD-7 and phq 8 were adopted in this study, although there are many established measures out there. Statistical analysis: Analyses used were robust and sound but the way how they are explained is insufficient. There were many analyses performed but some of them were not explained in this section. Therefore a table summarising analyses used is necessary. Results: descriptive and t-test, as well as logistic regression results reporting, is clear and informative but findings, especially for the latter, can be further improved particularly when the authors tried to explain how those factors are associated with both anxiety and depression. It seems to me the authors only emphasised anxiety rather than depression although both logistic results were displayed (please see 13-15). Therefore, considering massive result reporting, I suggest subheading under this result section is necessary to increase readability. Discussion: it is sufficient since some basic compare and contrast with previous literature were performed. However, it can be further improved especially to provide some reasoning as to why and how associations between those variables occurred? even they may be presumptive ones. Conclusion: Existing one is too short, thus i suggest that the limitation and strengths can be combined as one under the
--

	conclusion section. Besides, policy implication and future studies need to be highlighted also. Others: please see S Table 32: it should be Bonferroni... - The reviewer provided a marked copy with additional comments. Please contact the publisher for full details.
--	--

VERSION 1 – AUTHOR RESPONSE

Reviewer: 1

Prof. Graham Thornicroft, institute of psychiatry Comments to the Author:

Review of: Impact of the COVID-19 Pandemic on anxiety and depression symptoms of young people in the Global South: evidence from a four-country cohort study

The strengths of this paper include

- Two long established cohorts of young people
- A sensible array of risk and protective factors were assessed
- Standardised scales were used to assess anxiety and depression
- Important data are presented here on prevalence rates, and their differences across countries

Reply – We thank Prof. Thornicroft for his kind feedback and have addressed his helpful suggestions as detailed below.

The paper could be improved by

- 1) Clarify how people ‘without phones or internet access’ took part in a phone survey

Reply – This is an issue specific to the Ethiopian context, where many YL participants do not have access to a mobile phone. Thus, we used local guides who visited each person without access outdoors near their home and provided sanitised mobile phones for their participation. We address this point in the second paragraph of the methods section (page 4). It is worth to note that local guides were used in the previous (in-person) survey rounds too to support the fieldworkers in locating the Young Lives respondents as needed.

- 2) Saying more about why no pre baseline data on anxiety or depression were available

Reply – The Young Lives survey is a multidisciplinary prospective cohort survey that originally was set up to examine the lives of children growing up in poverty (added reference No. 29, page 4). The 2002 questionnaires have evolved over time to include relevant information, and the current team (author list) made the decision to introduce mental health measurement in 2020, given the likely effects of the pandemic. While the Young Lives project has measured mental health in the early survey rounds, for example through the emotional problem scale (EPS) a sub-scale of the Strengths and Difficulties Questionnaire (SDQ), the project had not specifically used the GAD-7 & PHQ-8 previously.

- 3) Discussing more about how far stigma related reluctance to disclose mental illness may at least in part account for the country difference in reported rates of anxiety and depression

Reply-We have searched the literature and we find evidence in all four countries about stigma and reluctance to disclose mental illness. These references (No. 64-69) are now provided in the text (page 16). However, we did not find any evidence to show that e.g. stigma in Peru should be less than in Vietnam, so we could not conclude that this is driving the difference in findings.

4) Saying why there was no patient or public involvement

Reply- There are no patients in the survey sample, it is a prospective cohort study, which does not have diagnostic research questions (see page 6). It is designed to be observational, and non-intervention is a requirement of this. One feature of the Young Lives survey which represents limited public involvement is “community reciprocity” where fieldworkers visit the communities and provide general information on the findings of the survey and get feedback, whilst preserving the anonymity of individual respondents.

5) Tidying up errors for reference insertion points eg on page 8 etc

Reply – We apologize for this error which we have fixed throughout the document.

6) I think that more needs to be said about the huge differences in prevalence rates eg between Peru and Vietnam, how can these be understood? Are differences in covid related mortality rates sufficient to explain the magnitude of these differences?

Reply – It is hard to overstate the difference in COVID mortality rates between Peru and Vietnam. Vietnam has recorded only 35 deaths in total (not per day), with no deaths since September 3rd, 2020. On October 15th (end date of our survey) Peru had registered 33,577 deaths. The population of Vietnam is around three times that of Peru. Another aspect that has likely contributed to the difference is the length of the lockdown, which creates stress and reduces household income. In Peru this was 107 days consecutive days at the national level, followed by an additional period of local lockdowns, such that certain areas of the country were in lockdown for up 199 days. In contrast, Vietnam had a very short and successful lockdown, one further localised lockdown, but by September life was already back to normal. We mention these aspects briefly in this revised version (see page 15).

7) What may explain why countries do or do not show gender differences in these prevalence rates?

Our prior was that women would show higher rates of anxiety and depression in general, e.g. McLean et al (2011) (added reference No. 60) p1 note “One of the most widely documented findings in psychiatric epidemiology is that women are significantly more likely than men to develop an anxiety disorder throughout the lifespan”. And Salk et al (2017) (added reference No. 61) note in a metaanalysis that larger gender differences were found in nations with greater gender equity, for major depression, but not depression symptoms. Furthermore, in the context of the pandemic, our prior is that mental health problems were likely to be larger among women given research on developed countries (e.g. Banks and Xu, 2020 for the UK). In three of the four countries, mental health is worse for females (either anxiety or depression or both). To us it was somewhat surprising that in Ethiopia there were no gender differences in the prevalence rates of either anxiety or depression. We have tried to investigate what is particular about the Ethiopia case, but there is very little evidence. In a pre-pandemic meta-analysis of mental distress in Ethiopian university students Dachew, et al. (added reference No. 62) find females to be more vulnerable. However, results from the Ethiopian National Health Survey (ENHS), which was conducted in nine Ethiopian regions, show no significant differences between males and females (added reference No. 63) which is in line with our findings. We have added the references in the discussion section (see page 16).

8) There appear to be 36 supplementary tables, are all these necessary?

Reply – Agreed that there are a lot of tables! But with four countries, two outcome variables, and two cohorts, the number of tables multiplies for each robustness check. We could have formatted these as

e.g. 4 long tables with panels for each country/gender/cohort but thought it was easier for the reader to access information through the separate tables if required.

Reviewer: 2

Dr. Julio Torales, National University of Asuncion Comments to the Author:

Very fine piece of work. I recommend that the authors add references that support the usefulness of using telephone surveys compared to "in-person" interviews.

Reply – We thank Dr. Torales for his kind words. The telephone survey was mainly employed to adhere to social distancing rules and travel restrictions which were present due to the COVID-19 pandemic. Oxford University applied an ethics policy of no face-to-face fieldwork during the pandemic therefore ethical approval for in-person interviews would not have been granted.

Reviewer: 3

Dr. Gabriel Teck, Universiti Teknologi Malaysia

*** This reviewer attached an annotated version of your manuscript alongside their review. You can find it attached to this email *** Comments to the Author:

- 1) Generally, this paper is of great importance and can be considered for publication after some revisions are performed. English is good and is grammatically sound except for some typos (please refer to the commented file- see the abstract some words are missing). Too many supplementary tables, although there are necessary. Perhaps reducing some of them or some can be embedded in the main texts.

Reply – Unfortunately, we are already at the limited number of tables/figures so we cannot embed any more to the main text. It is acknowledged that there are a lot of tables, but with four countries, two outcome variables, and two cohorts, the number of tables multiplies for each robustness check. We could have had long tables with panels for each country/gender/cohort but thought it was easier for the reader to access information through the separate tables in an appendix if they are interested.

- 2) Abstract: it is well summarised but it can be improved by adding in some important points, especially for the conclusion part.

Reply- We have rewritten the conclusion within the 300 word limit to reemphasise pandemic-related stressors more (page 2).

2a) Design: o perhaps inferential analysis techniques should be stated too

Reply – We now mention the logistic regression estimation strategy in the Design section of the abstract (page 2).

o please check the language...something is missing for the period.: A phone survey implemented August-October 2020 to

Reply –We apologize for this oversight and have changed the language in the first sentence of the Design sub-section in the abstract (page 2). o check the font type and size: Non-inclusion was due to nonlocation

Reply –We apologize for this oversight and have changed the font size to 11 Pt. and the font type to Calibri (page 2).

- Conclusion: o The conclusion presented emphasised the descriptive part of the findings.

Perhaps key association between the variables can be highlighted (at least the most prominent ones)

Reply – The conclusion in the abstract now includes more reference to COVID-19 stressors (page 2).

3) Introduction: good problem statement and research gap is well justified. But why only 4 countries and why those countries were selected? why these four? it seems the authors selected the countries arbitrarily?

Reply – We have changed “to participants of the Young Lives study “ to “as part of the Young Lives study” to clarify that we did not choose these four countries for this project specifically (see page 4), we have interviewed everyone from the study that was available. The phone survey was conducted with the Young Lives respondents, the same sample was defined back in 2001. The choice to focus the Young Lives study on Ethiopia, India, Peru and Vietnam was made in the early 2000s (see Barnett et al, 2013 for a description of the prospective cohort study design, added reference No. 29). We have clarified this further in footnote on page 4. Young Lives was designed to monitor the effectiveness of the Millennium Development Goals (2000-2015) in reducing childhood poverty in varied political-economic and socio-cultural settings. This led to the selection of four study countries, one from each of the main regions in the global south. Countries were chosen to allow cross-country comparison in children’s outcomes across a range of criteria, including levels of economic development, different political formations and degrees of exposure to life-threatening conditions such as extreme weather events and conflict. Also, the choice of the countries was opportunistic in the way it exploits existing connections between the original Young Lives team (based at the University of Reading and the London School of Tropical Hygiene and Medicine). Independent of the rationale behind the original country choice, we do consider these countries most helpful in understanding young people’s mental health in LMIC’s due to their geographical coverage and diverse COVID-19 experience and responses. For example, one third of the world’s youth lives in India.

4a) Perhaps, prior to methods, the literature review or introduction should highlight a theoretical framework providing clearer assumptions or directions of those factors/variables effects towards anxiety and depression levels.

Reply – Please see below (comments relate to similar issues).

4b) Prior to methods, i would suggest the authors to include a theoretical framework (preferably a diagram), consisting of some hypotheses especially for the purpose of inferential or association analyses later. This helps readers understand better the directions and assumptions used in this study.

Reply – Thank you for these comments– we have now refined our hypotheses that we consider as a broad theoretical framework, which is a set of three hypothesized correlates of anxiety and depression relating to 1) the macro/country environment (level of infection/death in the country of residence) 2) Pandemic related stressors (economic shocks, family illness, change in education/work etc) and 3) Individual, household and contextual structural background characteristics. We also include proxybaseline information as a robustness check. Our data and methods allow us to discuss the first hypothesis and to directly test the second and third hypotheses (see page 3-4). We have also restructured our tables and the results section to better reflect our theoretical framework (pages 6-7, 914 and Supplementary tables/figures) and have added a diagram on the theoretical framework for clarification as requested (Figure 2, new submission).

5) Some references can be added (please see the file attached).

- Within each country, hypothesised stressors related to changes in circumstances/behaviours/wellbeing

- some references are needed
- For economic and some other factors, please also see the recent publication by Ling,

G.H.T.; Md Suhud, N.A.b.; Leng, P.C.; Yeo, L.B.; Cheng, C.T.; Ahmad, M.H.H.; Ak Matusin, A.M.R. Factors Influencing Asia-Pacific Countries’ Success Level in Curbing COVID-19: A Review Using a Social–Ecological System (SES) Framework. Int. J.

Environ. Res. Public Health 2021, 18, 1704. <https://doi.org/10.3390/ijerph18041704> o Please also see this: Al Omari, O., Al Sabei, S., Al Rawajfah, O., Abu Sharour, L., Aljohani, K., Alomari, K., ... & Alhalaiqa, F. (2020). Prevalence and predictors of depression, anxiety, and stress among youth at the time of COVID-19: An online cross-sectional multicountry study. *Depression research and treatment*, 2020. o Khademian, F., Delavari, S., Koohjani, Z., & Khademian, Z. (2021). An investigation of depression, anxiety, and stress and its relating factors during COVID-19 pandemic in Iran. *BMC public health*, 21(1), 1-7.

- Similarly, depression and anxiety symptoms ... o same goes to this, references are needed.
Reply- Thank you for the reference suggestions they have been added (No. 19-21, page 3). We have further added reference No. 22-24 to support our hypothetical framework (page 4).

6a) Methods: Why PHQ8 ? why not Phq-9? some rationales should be provided as to why GAD-7 and phq 8 were adopted in this study, although there are many established measures out there.

Reply – After a review of the literature and consultation with the Oxford University Department of Psychiatry we decided to pilot the PHQ-9 and GAD-7 for in-person fieldwork because:

1. they are two standardised tools commonly used as diagnostic tools in observational non-clinical studies,
2. They have been validated in most of the Young Lives study countries/languages.

When we switched to a phone survey, we did cut PHQ-9 to PHQ-8 for ethical reasons (removing the question about suicidal thoughts):

1. It was deemed not feasible to deal with a situation where suicidal thoughts emerge from the interview (given that YL is not clinical-based and is not an intervention study). In that case, we would have had to make appropriate provision for people endorsing the suicidal thoughts question, however scarce availability of local services in case of needs made this a very difficult task, especially in the context of a pandemic (for instance, community mental health centres in Peru remained close during 2020, appointments were only available by phone).
2. We considered that it was too distressing to ask those type of questions over the phone, and it would also be difficult to notice non-verbal cues about distress.
3. Fieldworkers were not trained (and was not time to train them) to deal with someone experiencing suicidal symptoms.

The two tools were pre-piloted in Ethiopia and Peru as part of the preparation work for the in person R6 and piloted for the 4 countries for the phone survey. We now mention that we drop the PHQ 9th question for ethical reasons in the text (page 4).

6b) it is unclear that why only 4 countries were involved in this study? based on the young lives study?

Reply – Please see our response above to point 3 (REV3).

7) Statistical analysis: Analyses used were robust and sound but the way how they are explained is insufficient. There were many analyses performed but some of them were not explained in this section. Therefore a table summarising analyses used is necessary.

-Furthermore, i noticed there were many statistical analyses used, aside from the primary logistic regression, including t test, pearson correlation for correlation between GAD and PHQ? I advise the authors to provide a table to summarise all analyses used, and which variables/items were involved in that.

-please also include the t-test analysis since it has been employed. After looking at Table 1, ttest was used.

Reply – We added that we employ t-test to compare subgroups at the beginning of the section on statistical analysis (page 5). Unfortunately, we do not have space to add another table. Furthermore, both reviewer 1 and reviewer 2 noted the lengths of the existing Appendix. All analysis using t-tests are performed on the dichotomous outcome variables at least mild anxiety and mild depression across different sets of control variables. The calculation of the outcome variables as well as the set of control variables are already explained in the “Methods” section. We use Pearson’s correlation coefficient with Bonferroni corrected p-values to investigate i) the relationship between GAD-7 and PHQ-8 raw scores and ii) the relationship between subjective well-being (SWB) and GAD-7 and PHQ-8 raw scores, which we now mention in a footnote on page 6. Note that the calculation of all three measures has been explained to the reader in the “Methods” section. We have added the information about the type of p-value correction to Supplementary table 34. We believe that the BMJ open readership will be familiar with the methodology for t-test and correlations.

8a) Results: descriptive and t-test, as well as logistic regression results reporting, is clear and informative but findings, especially for the latter, can be further improved particularly when the authors tried to explain how those factors are associated with both anxiety and depression. It seems to me the authors only emphasised anxiety rather than depression although both logistic results were displayed (please see 13-15). Therefore, considering massive result reporting, I suggest subheading under this result section is necessary to increase readability.

-Good results but the reporting manner is confusing and unstructured.

-Or these findings mainly report the anxiety part? how about depression?

-a bit confusing...whether the authors were discussing about anxiety or depression? maybe the authors can have a separate section while maintaining those factors and stressors for both DVs

Reply – Indeed, given the word limit, we note on page 6-7 that given the similarity of the results, and space limitations, our discussion focuses on the anxiety results, and then at the end we note where these results differ from the depression results. It has been very challenging to summarise the results for four countries though we believe it is important to present all together in one paper. To further clarify this structure we have added the subheading “Logistic regression results (odds ratios): At least mild anxiety” on page 13. Here, we have omitted all references to at least mild depression in the paragraph on results for at least mild anxiety. As done in our original submission, we note differences between the regression results under the subheading “Significant differences between anxiety and depression logistic regression results” on page 14. Additionally, we now list all tables which include results referring to i) at least mild depression logistic regression outcomes, and ii) the relationship between SWB and depressive symptoms in this paragraph page (14).

8d) sub heading may be necessary for each finding or factor under the result section.

Reply – Please note that our original submission already contains subheadings for the different factors as set up in our theoretical framework (hypothesis).

8c) Please correct it

Reply – We apologize for this error which we have fixed throughout the document.

8d) Table 1: for the percentage, it should be 17.87 not the ones shown in the table. the dot should be at the lower part.

Reply – We apologize for this oversight and have lowered the decimal point in table 1 (page 8).

8e) Please rectify it

Reply – We apologize for this error which we have fixed throughout the document.

9) Discussion: it is sufficient since some basic compare and contrast with previous literature were performed. However, it can be further improved especially to provide some

reasoning as to why and how associations between those variables occurred? even they may be presumptive ones.

Reply – We have gone through the discussion to make it as clear as possible. The main issue is the word limit, so there is little room to add more discussion, so we did not prioritise presumptive or speculative discussion.

10) Conclusion: Existing one is too short, thus i suggest that the limitation and strengths can be combined as one under the conclusion section. Besides, policy implication and future studies need to be highlighted also.

-perhaps strengths and limitations can be part of the conclusion. Please also highlight future studies and some policy implications, aside from contributing new knowledge to fill the research gap.

Reply – Please note we followed the BMJ Open format to include the limitation and strengths separately and before the conclusion.

We now offer two potential policy suggestions in the conclusion. First, the expansion of telephone helplines, and second the expansion of Cash Transfer Programmes (CTPs) to help break the vicious cycle between poverty and mental illness. Further, we suggest a future country specific study in the form of lockdown and COVID-19 severity in Peru (see page 17).

11) Figure 1:

-please provide a reference

Reply – Please note that we provided references for this figure in the original submission. The respective references are 52 and 53 (original submission) and 71 and 59 (current submission).

12) Figure 2:

-this figure is quite interesting dividing rounds of surveys carried out for two cohorts using different set of factors or stressors for respective ages. Methodologically this is sound. But for better clarity or understanding of the study's primary work, a simple or more directly theoretical framework involving both IVs and DV (GAD-7 and PHQ-8) can be provided.

Reply –Thank you for the suggestion. We have now provided a graphic depiction of the three main hypotheses of our theoretical framework (Figure 2, current submission), including both independent variables and mental health as an outcome variable. Figure 2 has been adapted to fit the wording established in the theoretical framework (4a & 4b (REV3) and is now included as supplementary figure 1 (given that we are only allowed to include 2 figures in the manuscript as per the journal regulations).

Please see our response to comments 4a & 4b (REV3) on how we have sketched the theoretical framework with the three main hypotheses in relation to figure 2 (current submission).

13) Others: please see S Table 32: it should be Bonferroni...

Reply – We apologize for this error which is now fixed (page 42 in Supplementary tables).

Additional References (in order of appearance in manuscript)

19. Ling GHT, Md Suhud NAB, Leng PC, et al. Factors Influencing Asia-Pacific Countries' Success Level in Curbing COVID-19: A Review Using a Social-Ecological System (SES) Framework. *Int J Environ Res Public Health* 2021;18(4) doi: 10.3390/ijerph18041704 [published Online First: 2021/02/14]
20. Al Omari O, Al Sabei S, Al Rawajfah O, et al. Prevalence and Predictors of Depression, Anxiety, and Stress among Youth at the Time of COVID-19: An Online Cross-Sectional Multicountry Study.

Depress Res Treat 2020;2020:8887727. doi: 10.1155/2020/8887727 [published Online First: 2020/10/17]

21. Khademian F, Delavari S, Koohjani Z, et al. An investigation of depression, anxiety, and stress and its relating factors during COVID-19 pandemic in Iran. *BMC Public Health* 2021;21(1):275. doi: 10.1186/s16089-021-10329-3 [published Online First: 2021/02/05]
22. Favara M, Freund R, Porter C, et al. Young lives, interrupted: Short-term effects of the COVID-19 pandemic on adolescents in low- and middle-income countries. *Covid Economics* 2021;67:17298.
23. Burki T. The indirect impact of COVID-19 on women. *The Lancet Infectious Diseases* 2020;20(8):904-910. doi: 10.1016/s1473-3099(20)30568-5
24. Reichelt M, Makovi K, Sargsyan A. The impact of COVID-19 on gender inequality in the labor market and gender-role attitudes. *European Societies* 2020;23(sup1):S228-S45. doi: 10.1080/14616696.2020.1823010
29. Barnett I, Ariana P, Petrou S, et al. Cohort profile: the Young Lives study. *Int J Epidemiol* 2013;42(3):701-8. doi: 10.1093/ije/dys082 [published Online First: 2012/05/24]
60. McLean CP, Asnaani A, Litz BT, et al. Gender differences in anxiety disorders: prevalence, course of illness, comorbidity and burden of illness. *J Psychiatr Res* 2011;45(8):1027-35. doi: 10.1016/j.jpsychires.2011.03.006 [published Online First: 2011/03/29]
61. Salk RH, Hyde JS, Abramson LY. Gender differences in depression in representative national samples: Meta-analyses of diagnoses and symptoms. *Psychol Bull* 2017;143(8):783-822. doi: 10.1037/bul0000102 [published Online First: 2017/04/28]
62. Dachew BA, Bifftu BB, Tiruneh BT, et al. Prevalence of mental distress and associated factors among university students in Ethiopia: a meta-analysis. *J Ment Health* 2019;1-8. doi: 10.1080/09638237.2019.1630717 [published Online First: 2019/06/30]
63. Hailemariam S, Tessema F, Asefa M, et al. The prevalence of depression and associated factors in Ethiopia: findings from the National Health Survey. *International Journal of Mental Health Systems* 2012;6(23)
64. Nguyen A. Cultural and social attitudes towards mental illness in Ho Chi Minh City, Vietnam. *Seattle University Research Journal*, Spring 2003;2:27–31.
65. Reta Y, Tesfaye M, Girma E, et al. Public Stigma against People with Mental Illness in Jimma Town, Southwest Ethiopia. *PLoS One* 2016;11(11):e0163103. doi: 10.1371/journal.pone.0163103 [published Online First: 2016/11/29]
66. Arriola-Vigo JA, Stovall JG, Moon TD, et al. Perceptions of Community Involvement in the Peruvian Mental Health Reform Process Among Clinicians and Policy-Makers: A Qualitative Study. *International Journal of Health Policy and Management* 2019;8(12):711-22. doi: 10.15171/ijhpm.2019.68
67. Gaiha SM, Taylor Salisbury T, Koschorke M, et al. Stigma associated with mental health problems among young people in India: a systematic review of magnitude, manifestations and recommendations. *BMC Psychiatry* 2020;20(1):538. doi: 10.1186/s12888-020-02937-x [published Online First: 2020/11/18]
68. Kane JC, Elafros MA, Murray SM, et al. A scoping review of health-related stigma outcomes for high-burden diseases in low- and middle-income countries. *BMC Med* 2019;17(1):17. doi: 10.1186/s12916-019-1250-8 [published Online First: 2019/02/16]
69. Mascayano F, Armijo JE, Yang LH. Addressing stigma relating to mental illness in low- and middle-income countries. *Front Psychiatry* 2015;6:38. doi: 10.3389/fpsy.2015.00038 [published Online First: 2015/03/31]

70. Bauer A, Garman E, McDaid D, et al. Integrating youth mental health into cash transfer programmes in response to the COVID-19 crisis in low-income and middle-income countries. *The Lancet Psychiatry* 2021 doi: 10.1016/s2215-0366(20)30382-5

VERSION 2 – REVIEW

REVIEWER	Teck, Gabriel Universiti Teknologi Malaysia
REVIEW RETURNED	22-Mar-2021
GENERAL COMMENTS	The authors have made substantial revisions to the paper. It is well done; therefore, I do not have any further comment. The paper is now acceptable. Congratulations.